# Morphogenesis of New Straits and Islands Originated in the European Arctic Since the 1980s

Wieslaw Ziaja *[ID] and Krzysztof Ostafin[ID]

Institute of Geography and Spatial Management, Jagiellonian University in Cracow, Gronostajowa 7, 30-387 Krakow, Poland; krzysztof.ostafin@uj.edu.pl

* Correspondence: wieslaw.ziaja@uj.edu.pl

**Abstract:** Several new islands and many islets have appeared in the European Arctic since the end of the 20th century due to glacial recession under climate warming. The specificity of the formation of each individual strait and island is shown in the paper (apart from its location and timing of its origin). Analysis of available maps and satellite images of all three European Arctic archipelagos, from different times since 1909–1910, was the main research method. There are three pathways of the morphogenesis of the new islands: (1) simultaneous recession of glaciers from both sides of a depression in bedrock being a potential strait (typical in Franz Josef Land), (2) uncovering a rocky hill (which protrudes from a depression in bedrock) from under a receding glacier, (3) recession of one glacier which had reached a rocky fragment of a coastline (e.g., headland or peninsula), being a potential new island, during a maximum extent of this glacier during the Little Ice Age (in the beginning of the 20th century). Additional straits and islands are currently at the stage of formation and will continue to form in the European Arctic in the case of further warming or stabilization of the current climate conditions.

**Keywords:** climate change; glacial recession; coastal landscape transformation; new Arctic islands and straits; European Arctic

## 1. Introduction

In the European Arctic, the Little Ice Age began not later than in the middle of the 17th century [1] and lasted until the end of the 19th century [2]. Since the 1980s, the whole Arctic has undergone the current climate warming [3]. In the study area (Barents Sea with adjacent archipelagos), this warming was "evidently ( . . . ) the greatest" in the Arctic [4]. Strong evidence exists to show that this warming led to glacial recession in the area after the Little Ice Age, i.e., in the 20th and 21st centuries [5–7]. Hence, we omit other references to these events and set to outline the main subject of interest.

In the period 1963–2017, a total of 34 new islands—each with an area 0.5 km$^2$ or more—appeared as a result of the fragmentation of their coasts due to the recession of tide-water glaciers in the Arctic. Each island has been accurately mapped. Most of the new islands appeared near Greenland but seven appeared in the European Arctic. Of course, the formation of a new strait is necessary for the appearance of a new island. Each new strait came into existence due to the ablation of a glacier or glaciers, which had filled an open depression (below sea level) in bedrock covered previously with ice. Additional islands are in the course of the formation [8].

In this paper, we wish to broaden our knowledge of the eight new European islands beyond their location and formation time. Seven of the islands have been examined and discussed by us previously [8]. The smallest of the islands taken into account had an area of 0.4 km$^2$ (Table 1). Hence, our objective in this paper is to show the specificity of the formation of each individual strait and island

in order to find similarities and differences in their morphogenesis. The term "morphogesis" is defined as a manner in which landforms and glaciers change, leading to the formation of a strait and an island.

**Table 1.** New Arctic islands. Location numbers are the same as in Figure 1.

| No. Location | Old Name | Area (km$^2$) | Centroid | Elev. (m) | Year or Period of Origin |
|---|---|---|---|---|---|
| 1. Franz Josef Land | Eva-Liv Island | 0.4 | 81°42′04″ N, 62°43′40″ E | 20 | 1950s–1985 |
| 2. Franz Josef Land | Northbrook Island | 18 | 79°57′50″ N, 50°11′57″ E | 308 | 1986 |
| 3. Franz Josef Land | Hall Island | 59 | 80°11′30″ N, 58°12′09″ E | 420 | 2016 |
| 4. Novaya Zemlya | Chernyshev Glacier | 0.9 | 76°01′48″ N, 60°48′08″ E | <100 | 2004 |
| 5. Novaya Zemlya | Tajsiya Glacier | 0.6 | 75°57′56″ N, 60°20′12″ E | <100 | 2014 |
| 6. Novaya Zemlya | Upor Headland | 6.5 | 75°46′03″ N, 58°40′29″ E | 242 | 1993–1994 |
| 7. Novaya Zemlya | South Vilkitskiy Glacier | 0.4 | 75°34′32″ N, 58°16′33″ E | <50 | 2010–2011 |
| 8. Svalbard | Blomstrandhalvøya | 16.3 | 78°58′47″ N, 12°04′50″ E | 385 | 1991–1994 |

## 2. Materials and Methods

The materials used in the analysis of glacial recession leading to the formation of new islands and straits were collected for the most part via a survey of available maps and satellite images of all three European Arctic archipelagos, maps and images from different time periods since 1909–1910. The following collections of digital maps and satellite images at scales ranging from 1:100,000 to 1:500,000 were examined and used in the analysis in this paper:

- Norwegian maps of Svalbard: TopoSvalbard (2019) [9]
- Russian maps of the Russian Arctic: Soviet military topographic maps 1:100,000 and 1:200,000 (2019) [10]
- Landsat from NASA, in: USGS Global Visualization Viewer (2019) [11] for 1973–2019: Landsat 1–3 MSS, Landsat 4–5 MSS, Landsat 4–5 TM, Landsat 7 SLC, Landsat 8 OLI
- Sentinel-2 from ESA, in: Copernicus Open Access Hub (2019) [12] for 2018 and 2019.

A total of 8 new islands—each occupying 0.4 km$^2$ or more—were discovered in this manner (Figure 1, Table 1), and their formation process was established via a figure containing at least three cartographic pictures—including one before the formation of the island (a map) and a second after its formation (a satellite image).

For Franz Josef Land and Novaya Zemlya we used the aforementioned Russian topographic maps created on the basis of aerial and geodetic surveys from 1952 [13,14] and published in the 1960s and 1970s. For Kongsfjorden in Svalbard, we used a topographic map (1:200,000) based on the Gunnar Isachsen expedition measurements performed in 1909–1910 [15]. These Russian and Norwegian maps were georeferenced in ArcMap 10.7, based on 10 to 20 common topographical points between satellite images and maps. The RMS error ranged between 10 and 30 m.

The second method of collecting data was a survey of the literature on new islands, mainly on the Internet. This literature consisted mostly of three kinds of popular science notes: (1) originally observed, denoted, and published, as indicated by Pelto (2009–2019) [16], as well as some compiled by Pelto (2017) [17] on the Novaya Zemlya and Spitsbergen coasts, and by Sharov (2014) [18], (2) other Internet pages without authors' names, and (3) Internet pages of press agencies with news.

Apart from analyzing maps, satellite images and literature, direct summer field investigations of new straits and islands were carried out by us in Spitsbergen. In 1995, one of the authors explored a new strait and island in Kongsfjorden, just after their formation. In 2005 and 2016, both authors carried out detailed landscape (including geomorphological) mapping of the coast near the glacial isthmus (formed of two glaciers under recession) being transformed into a new sound between the Sørkapp Land peninsula and the rest of Spitsbergen [5,8,19].

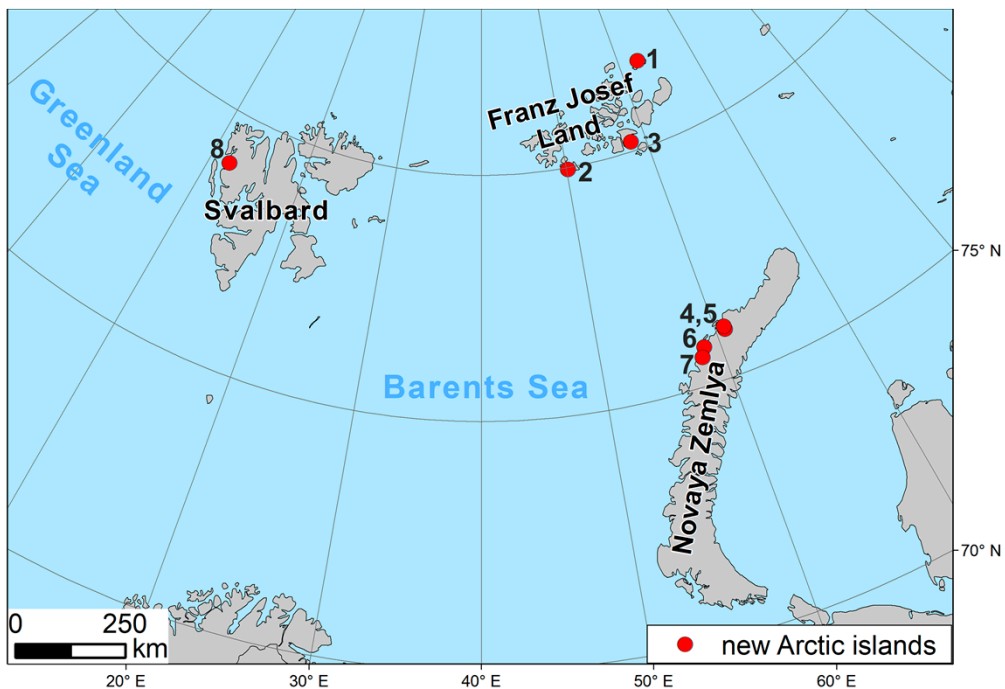

**Figure 1.** Locations of new islands in the European Arctic; location numbers are the same as in Table 1.

## 3. Results: Analysis of New Straits and Islands

### 3.1. New Island—Former Mesyatsev Headland of Eva-Liv Island—Franz Josef Land

The former Mesyatsev Headland—the northwesternmost part of Eva-Liv Island which is the northeasternmost part of the Franz Josef Land archipelago—was transformed into a small new island [8,18] after the formation of a new strait due to the recession of two ice sheets in the west of Eva-Liv Island. This must have happened before August 2nd, 1985. On this day, the land area below 300 m a.s.l.—i.e., both ice sheets in the west and the majority of the main eastern ice sheet (apart from its top)—remained free of snow. It is interesting that the surface of the new island decreased severalfold from 1985 to 2018 (Figure 2). In the case of further climate warming, the process of the island's decline will continue and the island will disappear in the nearest future if its bedrock becomes situated below sea level.

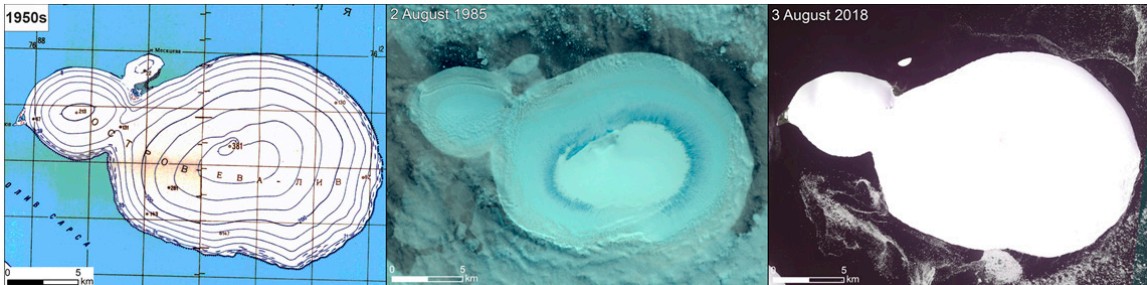

**Figure 2.** Transformation of the Mesyatsev Headland (of Eva-Liv Island, Franz Josef Land) to a new island.

Environmental conditions including bedrock and climate determined the disappearance of two other islands in Franz Josef Land (which were small and similar to the aforementioned one) in the nearest past. Sharov (2005) [13] evidenced that Perlamutrovy (Pearl) Island, near the southern coast of Graham Bell Island, and the largest of the Lyuriki Islands found near the southern coast of Mc Clintock Island were completely overtaken by the sea. The first of the islands disappeared before 1984 (Figure 3).

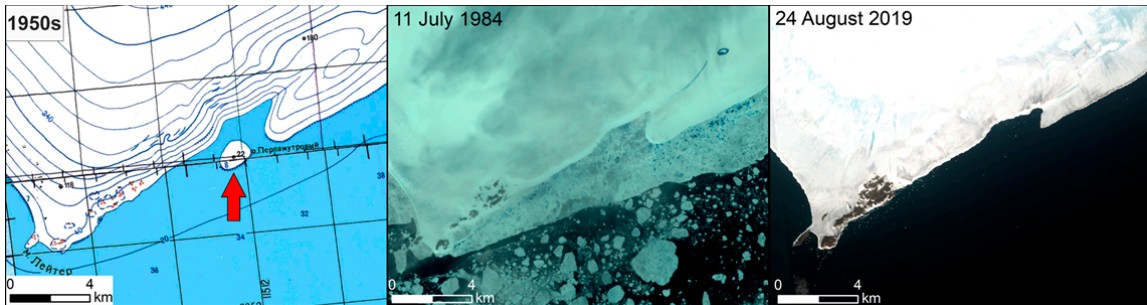

**Figure 3.** Disappearance of Perlamutrovy Island near the southern coast of Graham Bell Island, Franz Josef Land.

*3.2. New Yuriy Kuchiev Island—Former West of Northbrook Island—Franz Josef Land*

The former Northbrook Island became divided into two parts due to the recession of glaciers which had filled the bedrock depression (in the north-south direction) in the west. According to RIA Novosti (2012) [20], the new strait and island, taking the westernmost part of former Northbrook Island, were discovered by an expedition of the Russian Navy in the summer 2012 and given a navy officer's name. This island's origin was not illustrated by any published map or picture. Hence, we made such a figure using, among other data, a Landsat image from 2006, which showed the new strait and island which occupies 18 km$^2$ [5,8] of surface area, at least 6 years before their discovery. However, afterwards, Landsat 4-5 MSS images from 1986 were found, which serve a proof that the new strait and island appeared 26 years before the discovery of this fact (Figure 4). The said strait formed due to recession of glaciers which had filled a depression in bedrock from both the east and west. It is certain that no shelf glacier existed there in the Little Ice Age and 20th century because the said glaciers were significantly declined and fissured, unlike in the case of shelf glaciers or land fastened multiannual sea ice which would have to be flat. Apart from that, the study area had been relatively often and extensively explored. However, it is quite strange that the new strait was not discovered by international summer expeditions in 1990–1992 [21]. This may have been caused by bad weather and/or sea ice between the two glacial coastlines of the strait; the least probable was a frontal re-advance of the glaciers which re-filled the strait's depression for such a short period then. However, in the 1950s, the glaciers have a clearly smaller extent on both sides of the strait. Since 1986, the new strait between the two studied islands has been partly filled in with marine deposits and thus narrowed (Figure 4).

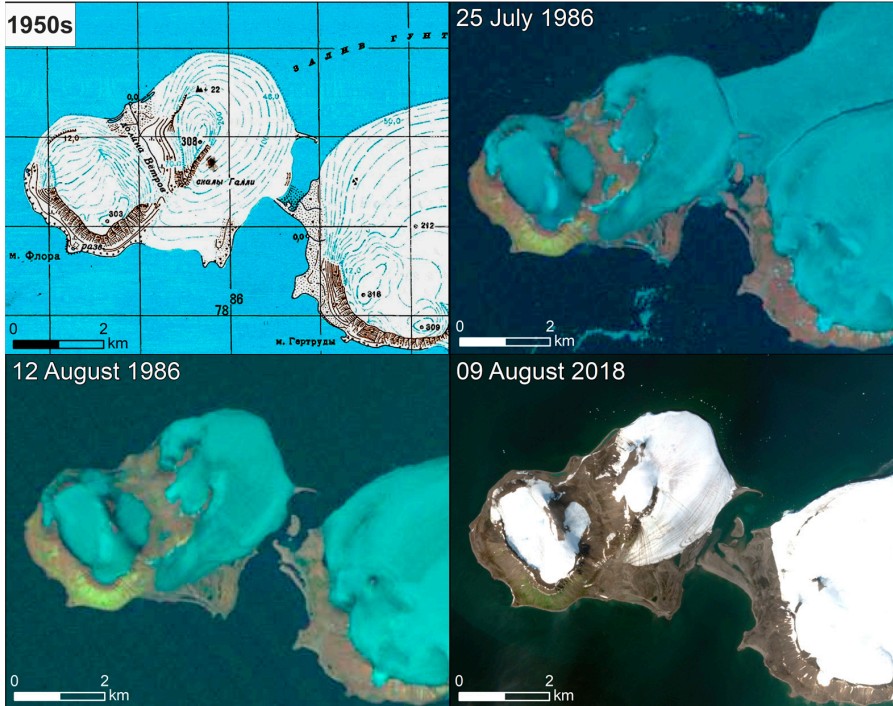

**Figure 4.** Transformation of the western part of Northbrook Island into the new Island of Yuriy Kuchiev.

### 3.3. New Island—Former Littrov Peninsula of Hall Island—Franz Josef Land

The eastern part of Hall Island became separated from its mainland between in 2016 [6] due to the same, as described above, process of progressive narrowing of the glacial connection between the former Littrov Peninsula and the rest of Hall Island. In fact, two ice (glacial) sheets—the bigger one in west and smaller one in the east—split due to recession, i.e., decreasing glacier thickness (lowering of their surface) under the summer ablation of ice (after ablation of snow cover on ice sheets). The process was attentively observed and described by Sharov and Zaprudnova (2014) [22] but their labeling of the glaciers' junction as an "ice bridge" is not correct because the strait depression had been filled by clearly declining and fissured slopes of ice sheets, and not by a flat shelf glacier or land-fastened sea ice (Figure 5). This island is the biggest—59 km$^2$ of surface area—of all new islands which have appeared in the European Arctic.

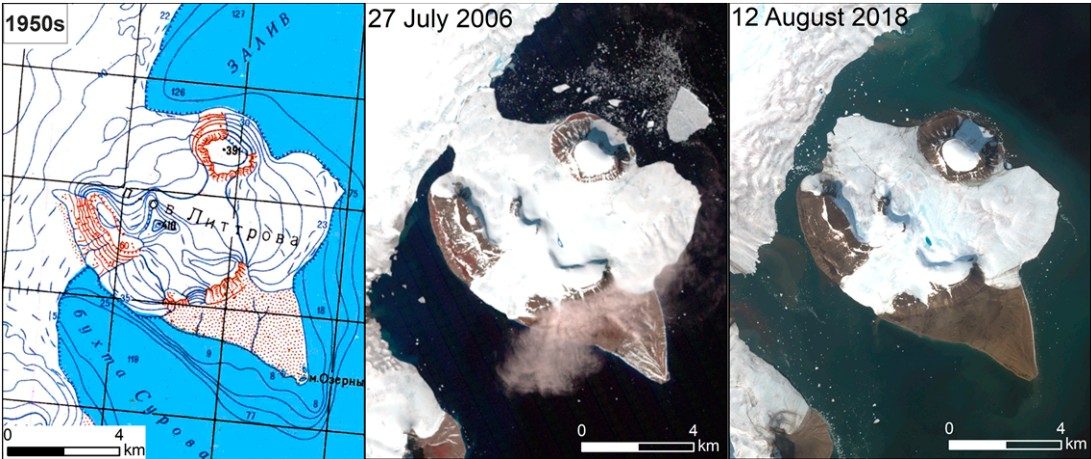

**Figure 5.** Transformation of the Littrov Peninsula (of Hall Island, Franz Josef Land) into a new island.

### 3.4. Two New Islands—North and South Coasts of the Pankratyev Peninsula—Novaya Zemlya

Two small new islands became freed from underneath tide-water glaciers on the northwestern coast of Novaya Zemlya, certainly after the year 2000, but before August 27th, 2014 or the day of the first Landsat image, which showed the new islands and straits. The bigger, northeastern island is situated more than 3 km from the ice cliff of Chernyshev Glacier, while the smaller, southeastern island ca. 2 km from the front of Tajsiya Glacier. Both tide-water glaciers are in a state of shrinkage since at least the 1970s. The two islands formed from sub-glacial rock hills surrounded by depressions in bedrock, covered with the thick ice-tongues of glaciers at the end of the Little Ice Age (beginning of the 20th century). Both the islands, together with a strait dividing them from the glacial coastline, appeared due to recession of only one glacier (Figure 6). The appearance of both these islands was evidenced by Pelto (2017) [17] as well as Ziaja and Ostafin (2019) [8].

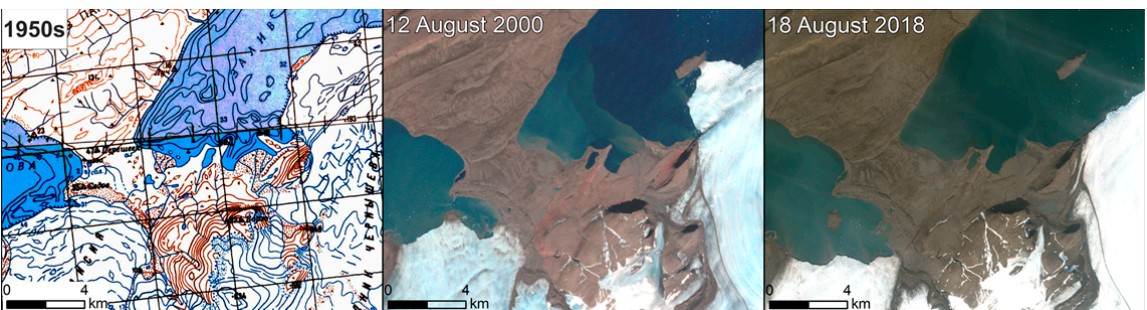

**Figure 6.** Uncovering of two small new islands from underneath the Tajsiya (left) and Chernyshev (right) glaciers on the coast, respectively south and north of the Pankratyev Peninsula, NW Novaya Zemlya.

### 3.5. New Island—Former Peninsula with Upor Headland—Novaya Zemlya

The recession of Krivoshein Glacier, found in the northwest of Novaya Zemlya, led to the formation of a strait which isolated the former peninsula with the Upor Headland from the glacier's ice cliff just before the year 1994. A relatively high elevation of the former peninsula above the glacier in 1971 and visible sites with continuous vegetation in 1994 evidence the location of this area outside of the maximum glaciers extent in the Little Ice Age (Figure 7). Hence, the new island was formed due to recession of only one glacier (contrary to the aforementioned islands of Franz Josef Land) and the island's surface area had not been completely covered by glaciers earlier (unlike the new small islands of Novaya Zemlya mentioned above and below). The appearance of this midsize island—6.5 km$^2$ in area—was described by Pelto (2017) [17] as well as Ziaja and Ostafin (2019) [8]. The new strait between the island and its mainland is 1 km wide, and the distance to the tide-water glacier front is 5 km.

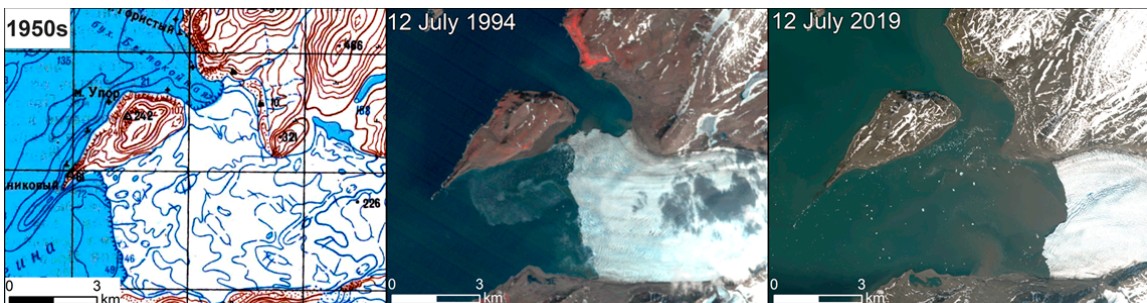

**Figure 7.** Transformation of the peninsula with Upor Headland to a new island, NW Novaya Zemlya.

### 3.6. New Islands—Front of South Vilkitskiy Glacier—Novaya Zemlya

Two new small rocky islands (and one islet in between) became uncovered from underneath South Vilkitskiy Glacier in the northwest of Novaya Zemlya after the year 2010 (Figure 8). According

to our interpretation of Landsat and Sentinel-2 images, the bigger of the islands, which occupies 0.4 km$^2$ of surface area, and shown in Table 1—became separated from the glacier in 2010–2011, while the smaller island—0.2 km$^2$ of surface area only—appeared in 2017–2018. The formation of the two islands was evidenced in 2017 by the Russian State Corporation "Roskosmos" [23] and Pelto [17]. The morphogenesis of these new islands and straits (and the glacier's cliff) is the same as in the case of a new island north and south of the Pankratyev Peninsula.

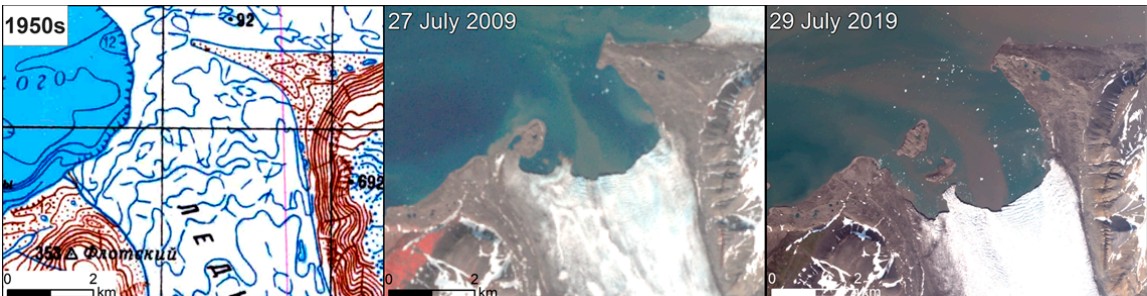

**Figure 8.** Uncovering of two small new islands and one islet from under South Vilkitskiy Glacier, NW Novaya Zemlya.

### 3.7. New Island–Former Bloomstrandhalvøya Peninsula–Svalbard

The only new island, apart from islets, on the Spitsbergen coast (the biggest island of the Svalbard archipelago), occupying 16.3 km$^2$ of surface area (Table 1), became transformed from the former Bloomstrandhalvøya peninsula in Kongsfjorden, as result of a decrease in the thickness of the Blomstrandbreen glacier [5]. Ablation of the ice surface of the glacier's tongue led to is decline in the area of the depression of bedrock, which became a new strait in 1991–1994. In 1995, one of the authors sailed through this strait by boat. As we can see, three new islets (not included in Table 1), appeared in the strait—two of them before the 1990s and the third afterward. Before the glacier's recession, the islets consisted of rock hills stretching from the bedrock depression above sea level (Figure 9). The pattern of formation of these islets is the same as in the case of the aforementioned former peninsula with the Upor Headland (in Novaya Zemlya) because it certain that its area had not been covered with glacier during the Little Ice Age. This is evidenced at a map from 1909–1910 [15].

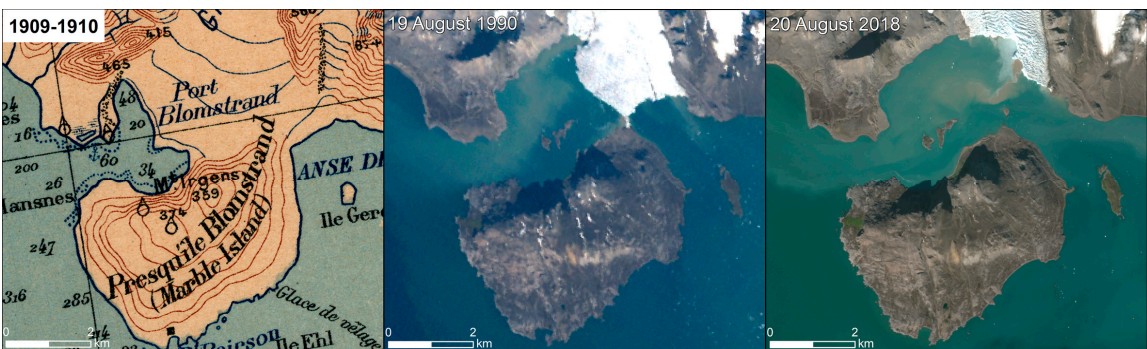

**Figure 9.** Transformation of the Blomstrandhalvøya peninsula into a new island in Kongsfjorden, NW Spitsbergen.

### 3.8. Numerous Small Islets in Many Different Areas

Apart from the aforementioned new islands, occupying 0.40 km$^2$ of surface area, and islets nearby, many small islets appeared at the fronts of recessing glaciers. They are mostly rocky, formed due to the transformations of small hills (protruding up from depressions) which had been covered with the glacier tongues before glacial recession, e.g., an islet occupying 0.06 km$^2$ of surface area on the eastern coast of Northbrook Island. However, some of the new islets are built of moraine and marine deposits

accumulated by glaciers or the sea, e.g., the former Morenetangen headland on the northern edge of Isbukta (eastern Sørkapp Land coast, SE Spitsbergen).

12 new islands were discovered by expeditions of the Russian Northern Navy along the coasts of Franz Josef Land and Novaya Zemlya since 2013 [24,25]. The biggest of the islets, ca. 0.13 km$^2$ in surface area, was uncovered from underneath Vize Glacier in the northwest of the north island of Novaya Zemlya at the beginning of 2016 [26,27].

## 4. Islands at the Stage of Formation

According to our observations on satellite images, there are also numerous potential islands, i.e., islands at the stage of formation.

The biggest of the potential islands, not only in the European Arctic but also in all the Arctic, is Sørkapp Land—the southern Spitsbergen peninsula occupying ca. 1300 km$^2$ of surface area. Its path towards a quick transformation into a new island, after the connection of two opposite fjords due to the recession of the glacial isthmus between them, was described by Ziaja and Ostafin (2015, 2019) [5,8] and shown by Grabiec et al. (2017) [28].

A small island is currently forming in the inner part of Kongsfjorden, in NW Spitsbergen. Two or three additional new small islands, also up to ca. 0.40 km$^2$ in surface area each, may appear due to further recession of the tide-water Vasilievbreen glacier's front (which was, in fact, a glacial piedmont cliff of many Sørkapp Land glaciers, now in the state of division) in SE Spitsbergen.

A new small island, ca. 0.40 km$^2$ in surface area, is now at the stage of formation along the northeastern coast of Northbrook Island (Franz Josef Land).

Two new islands are being uncovered from underneath the front of Krayniy Glacier along the northwestern coast of Novaya Zemlya (its northern island) between the Pankratyev Peninsula in the north and the new island with the Upor Headland in the south—as shown by Pelto (2017) [17] who also provides a few other examples of such a process in Novaya Zemlya.

## 5. Discussion

The formation of the new straits and thus islands is obviously conditioned by the recession of tide-water glaciers which had filled these straits' depressions (below the sea level) in bedrock and covered, partly or completely, these bedrock elevations (above the sea level), which became the new islands in question. Hence, the assertion that within "the ice cover of the north island of Novaya Zemlya ( . . . ) the position of ice margins has remained unchanged over the last 78 years", made by Ehlers et al. (2015) [1], may be true of land-terminating glaciers but it is not true for tide-water glaciers, at least in the aforementioned areas with new islands and straits formed since the end of the 20th century and in the beginning of the 21st century (Figures 6–8).

Moreover, citing Zeeberg and Forman (2001) [29] and Zeeberg (2002) [7] as a source of this assertion is not convincing because a significant retreat of tide-water glaciers in and around northwest Novaya Zemlya, in the Barents Sea, is stated in both these publications. Certainly, in both publications, it is evidenced that the eastern extent of land glaciers of the northern ice cap of Novaya Zemlya, from the Kara Seaside, did not change significantly after the Little Ice Age during the 20th century. According to Zeeberg (2002) [7], "Novaya Zemlya's land-based northeastern ice margin, extending over a distance of 150 km, varied little between 1952 and 1993 and in places is inset behind 'Little Ice Age' moraines". However, this does not refer to the tide-water glaciers of the northwest.

In addition, the tide-water glaciers described by them [7,29] are not the same as the glaciers which have produced new straits and islands due to their recession, as noted in the present study.

According to Sun et al. (2016) [30], the melting of tide-water glaciers was "the most significant contributor to the ice loss" in Novaya Zemlya, in the 21st century. Also, the new glaciers inventory for Novaya Zemlya [31] showed that a bigger area has been abandoned by glaciers than transgressed by them.

Melkonian et al. (2016) [32] reported general ice loss from the glaciers of Novaya Zemlya basing on their newest satellite data analyses. Also, in Franz Josef land glacier retreat is widespread and in general a trend of increasing glacier thinning is observed from the NE towards the SW (Zheng et al. 2018) [33]. The process of glacial recession is even more intensive in Svalbard [5,8,19]. Hence, subsequent coastal areas will become transformed into new islands and straits within the studied archipelagos.

## 6. Summary and Conclusions

The described change patterns which have altered significantly the map of the Arctic are being observed and described by a comparatively small number of authors: Pelto [16,17], Sharov with collaborators [13,18,22], as well as Ziaja and Ostafin [5,8,19].

There are three morphogenesis pathways of the new islands:

(1) simultaneous recession of glaciers (including ice sheets) from both sides of a depression in bedrock being a potential strait—typical in Franz Josef Land,
(2) uncovering a rocky hill (which protrudes from a depression in bedrock) from underneath a receding glacier—often in Novaya Zemlya and Svalbard,
(3) recession of one glacier which had reached a rocky fragment of a coastline (e.g., headland or peninsula), being a potential new island, during a maximum extent of this glacier in the Little Ice Age (in the beginning of the 20th century)—in Svalbard and Novaya Zemlya.

The last two morphogenesis pathways may be combined into a single process in a glacier recession leading to a new island's formation since the end of the Little Ice Age. This has often happened in Greenland [8]. However, this has not yet happened in the European Arctic.

New straits and islands are at the stage of formation and will be formed in the European Arctic in the event of further warming or stabilization of current climate conditions. This process will be become reversed in the event of climate cooling.

**Author Contributions:** Individual contributions of both authors are the same (50% each). Nevertheless, W.Z. was responsible for the text and K.O. was responsible for the figures.

**Funding:** This research received no external funding.

**Acknowledgments:** The two anonymous reviewers are thanked for their notes which enabled us to improve the paper's content. G. Zebik from the Jagiellonian University in Poland is thanked for his assistance in proofreading the text, and A. Psomas from the Swiss Federal Institute for Forest, Snow and Landscape Research is thanked for consultation.

**Conflicts of Interest:** The authors declare no conflict of interest.

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
