# Peer review of "Morphogenesis of New Straits and Islands Originated in the European Arctic Since the 1980s"

_geosciences, doi:10.3390/geosciences9110476_

Round 1

Reviewer 1 Report

The authors took into account my comments and suggestions. The manuscript has been greatly improved and now deserves publication in the journal Geosciences.

Author Response

Dear Reviewer 1,

Thank you very much for your quick work and high estimation of the revised version of our paper.

Best regards,

Wieslaw Ziaja & Krzysztof Ostafin

Reviewer 2 Report

The authors have made the necessary revisions and the paper is now improved and almost ready for publication. I have just one query regrading one of the revisions in the Discussion.

In your new discussion text you state:

Hence, the assertion that within “the ice cover of the north island of Novaya Zemlya (…) the position of ice margins has remained unchanged over the last 78 years”, made by Ehlers et al. (2015) [1], is not true, at least in the aforementioned areas with new islands and straits formed since the end of the 20th century and in the beginning of the 21st century (Figs. 6–8).

This is a little misleading since you do not include the rest of the Ehlers text for context, which is referring mainly to land-based glacier margins and later cites the supporting papers of Zeeber & Forman and Zeeberg. Your point is largely in relation to tide water glaciers, which is fair, as you note later in the Discussion. So the assertion of Ehlers et al. is still true for land-based glaciers whose margins have changed little over the last 78 years. It is the tide water glaciers that have changed a lot, as shown in your study.

I would recommend keeping your comments but changing the wording slightly to be fairer and more honest in relation to the original source that you are quoting: “the ice cover of the north island of Novaya Zemlya (…) the position of ice margins has remained unchanged over the last 78 years”, made by Ehlers et al. (2015) [1], may be true of land-terminating  glaciers but it is not true for tide-water glaciers, at least in the aforementioned areas with new islands and straits formed since the end of the 20th century and in the beginning of the 21st century (Figs. 6–8).

Author Response

Dear Reviewer 2,

Thank you very much for your quick work and positive estimation of the revised version of our paper. Your suggestion was realized by us in the form exactly the same as you wished.

Best regards,

Wieslaw Ziaja and Krzysztof Ostafin